# Mechanistic Interpretability of Code Correctness in LLMs via Sparse Autoencoders

## Abstract

As Large Language Models become integral to software development, with substantial portions of AI-suggested code entering production, understanding their internal correctness mechanisms becomes critical for safe deployment. We apply sparse autoencoders to decompose LLM representations, identifying directions that correspond to code correctness. We select predictor directions using t-statistics and steering directions through separation scores from base model representations, then analyze their mechanistic properties through steering, attention analysis, and weight orthogonalization. We find that code correctness directions in LLMs reliably predict incorrect code, while correction capabilities, though statistically significant, involve tradeoffs between fixing errors and preserving correct code. Mechanistically, successful code generation depends on attending to test cases rather than problem descriptions. Moreover, directions identified in base models retain their effectiveness after instruction-tuning, suggesting code correctness mechanisms learned during pre-training are repurposed during fine-tuning. Our mechanistic insights suggest three practical applications: prompting strategies should prioritize test examples over elaborate problem descriptions, predictor directions can serve as error alarms for developer review, and these same predictors can guide selective steering, intervening only when errors are anticipated to prevent the 14.66% corruption rate from constant steering.

## 1 Introduction

Large language models have achieved substantial adoption in software development, with 30% of GitHub Copilot's suggestions across one million developers entering production (Dohmke et al., 2023). Yet these same models fail on bug-prone code, achieving only 12.27% accuracy while reproducing 44% of historical training bugs verbatim (Guo et al., 2025). This contradiction between widespread deployment and fundamental reliability issues poses critical risks in healthcare, finance, and military applications where code failures have severe consequences. The core challenge lies in our lack of mechanistic understanding of how models internally determine code validity. While current code interpretability research provides insights, these approaches cannot isolate the specific mechanisms that distinguish when models will generate correct code versus reproduce training errors, limiting our ability to ensure reliable deployment.

Understanding code correctness mechanisms requires isolating specific features within model representations, yet neural networks complicate this through superposition, compressing thousands of features into fewer dimensions (Elhage et al., 2022). This compression creates polysemantic neurons that respond to completely unrelated concepts: a single neuron might fire for Python syntax, academic citations, HTTP requests, and Korean text simultaneously (Bricken et al., 2023). Current code interpretability research, while revealing what models encode about code structure, does not address this fundamental challenge of feature entanglement (Troshin & Chirkova, 2022; Paltenghi et al., 2024; Anand et al., 2024). Sparse autoencoders provide a solution by decomposing these superposed representations into interpretable components, having already proven effective at isolating mechanisms for entity recognition (Ferrando et al., 2024) and safety-relevant behaviors (Templeton et al., 2024) in natural language. Applying this decomposition to code generation represents an unexplored opportunity to determine whether models possess code correctness mechanisms.

Inspired by Ferrando et al. (2024)'s success in identifying entity recognition and uncertainty directions through SAEs, we adapt their framework to uncover how models internally represent code correctness. We apply their methodology to Gemma-2 using MBPP problems, analyzing residual streams at final prompt tokens where semantic information concentrates across layers (Geva et al., 2023; Lieberum et al., 2024). This suggests models employ two distinct mechanisms for code validity, with detection directions (identified via t-statistics) serving as error alarms and steering directions (via separation scores) enabling targeted corrections.

Overall, our contributions are as follows:

- Using sparse autoencoders, we discover **detection directions predict errors reliably** (F1: 0.821) but fail as confidence indicators for correct code (F1: 0.504), revealing asymmetry in how models represent code correctness.

- Through steering interventions, we show **steering interventions produce significant corrections while introducing tradeoffs**, fixing 4.04% of errors but corrupting 14.66% of correct code, necessitating selective rather than universal application.

- Attention analysis reveals **test cases matter more than problem descriptions** for both mechanisms, with correct-steering increasing test attention (+14.60) and incorrect-steering suppressing it (-12.69), while both ignore problem text.

- Weight orthogonalization proves **correct directions are necessary for generation**, their removal causing 83.62% corruption versus 18.97% for control.

- We demonstrate **code correctness features persist from base to chat models**, with incorrect-predicting and correct-steering directions from base Gemma-2 retaining their effectiveness after instruction-tuning, despite SAEs being trained only on base model representations.

## 2 SPARSE AUTOENCODERS

Sparse autoencoders are motivated by the Linear Representation Hypothesis, which posits that neural networks encode meaningful concepts as directions in activation space (Park et al., 2023; Mikolov et al., 2013). However, models face a fundamental challenge: they must compress thousands of features in lower-dimensional spaces, creating superposition where individual neurons respond to multiple unrelated concepts (Elhage et al., 2022). A single neuron might simultaneously activate for Python syntax, academic citations, HTTP requests, and Korean text (Bricken et al., 2023). This polysemantic behavior makes it difficult to isolate specific mechanisms like code correctness detection from the entangled representations.

Dictionary learning provides a principled solution to decompose these superposed representations (Olshausen & Field, 1997). We employ pre-trained JumpReLU SAEs from GemmaScope (Lieberum et al., 2024), which expand model representations $\mathbf{x} \in \mathbb{R}^d$ into an 8× larger sparse space $a(\mathbf{x}) \in \mathbb{R}^{d_{\text{SAE}}}$, enabling fine-grained feature separation. The encoding process applies:

$$a(\mathbf{x}) = \text{JumpReLU}_\theta(\mathbf{x}\mathbf{W}_{\text{enc}} + \mathbf{b}_{\text{enc}}) \tag{1}$$

where JumpReLU implements threshold activation:

$$\text{JumpReLU}_\theta(\mathbf{x}) = \mathbf{x} \odot H(\mathbf{x} - \theta) \tag{2}$$

Here, $H$ represents the Heaviside step function and $\theta$ is a learnable threshold vector. The decoder reconstructs through

$$\text{SAE}(\mathbf{x}) = a(\mathbf{x})\mathbf{W}_{\text{dec}} + \mathbf{b}_{\text{dec}} \tag{3}$$

Training optimizes a combined loss that balances reconstruction accuracy with sparsity:

$$\mathcal{L}(\mathbf{x}) = \underbrace{\|\mathbf{x} - \text{SAE}(\mathbf{x})\|_2^2}_{\mathcal{L}_{\text{reconstruction}}} + \underbrace{\lambda\|a(\mathbf{x})\|_0}_{\mathcal{L}_{\text{sparsity}}} \tag{4}$$

This encourages monosemantic features where each dimension captures a single interpretable feature (Bricken et al., 2023; Templeton et al., 2024). We refer to $a_i(\mathbf{x})$ as **latent activations** (feature presence strength) and $\mathbf{W}_{\text{dec}}[:, i]$ as **latent directions** (learned feature vectors) throughout our analysis.

## 3 RELATED WORK

Recent advances in mechanistic interpretability have mapped how language models process factual information, including entity recognition mechanisms (Ferrando et al., 2024), specialized extraction heads that route attributes to final positions (Geva et al., 2023), and structured circuits for factual recall (Nanda et al., 2023). These discoveries reveal interpretable, causal mechanisms underlying natural language processing. However, equivalent mechanistic understanding for code generation remains limited.

Current code interpretability research has provided valuable insights through probing classifiers, attention analysis, and model representation studies (Troshin & Chirkova, 2022; Paltenghi et al., 2024; Anand et al., 2024). While these approaches reveal what models encode about code structure and how they process information, they do not address the fundamental challenge of superposition—where models compress multiple features into the same neurons, making individual features difficult to isolate and interpret (Elhage et al., 2022). This superposition phenomenon means that analyzing raw activations provides entangled, polysemantic signals rather than clean, interpretable features.

Sparse autoencoders offer a solution by decomposing these entangled representations into interpretable components. SAEs have identified safety-relevant features (Templeton et al., 2024) and demonstrated their ability to extract meaningful computational structures from complex representations. Our work applies this methodology to code generation, employing validation through activation steering (Turner et al., 2023), weight orthogonalization (Arditi et al., 2024), and attention analysis (Nanda et al., 2023; Geva et al., 2023). This represents the first application of sparse autoencoders to address superposition in code representations, extending entity recognition methodologies to uncover how models internally represent program validity.

## 4 METHODOLOGY

We identify code correctness mechanisms in the Gemma-2-2b base model (Team et al., 2024) by extending Ferrando et al. (2024)'s entity recognition framework. Our approach discovers complementary mechanisms: detection directions that signal errors through confidence gradients, and steering directions that enable corrections through categorical activation patterns.

Using Mostly Basic Python Problems (MBPP) (Austin et al., 2021), a benchmark of 1,000 Python problems with test-based evaluation, we generate binary-labeled samples via the pass@1 criterion with temperature 0 for deterministic outputs. Each prompt is formatted using a standardized template (Figure 1) containing three components: problem description, test cases, and a code initiator. We add the code initiator to guide the base model to start generating code. Passing all three test cases yields correct labels; any failure produces incorrect labels. Data splits allocate 50% for direction selection, 10% for threshold & coefficient calibration, and 40% for mechanistic analysis.

```
    Problem      Write a function to find the minimum cost path to reach (m, n) from (0,
  Description    0) for the given cost matrix cost[][] and a position (m, n) in cost[][].

  Test Cases     "assert min_cost([[1, 2, 3], [4, 8, 2], [1, 5, 3]], 2, 2) == 8",
                 "assert min_cost([[2, 3, 4], [5, 9, 3], [2, 6, 4]], 2, 2) == 12",
                 "assert min_cost([[3, 4, 5], [6, 10, 4], [3, 7, 5]], 2, 2) == 16"

Code Initiator   # Solution:
```

Figure 1: Standardized prompt template for MBPP problems containing problem description, test cases with function signatures, and code initiator.

We extract residual stream activations at the final token before the model's answer begins, following Marks & Tegmark (2023) who demonstrated that end-of-instruction tokens aggregate information about the entire question. This approach, also employed by Ferrando et al. (2024) for uncertainty directions, captures the model's complete understanding of the problem specification before generation commences. Pre-trained GemmaScope autoencoders (Lieberum et al., 2024) decompose these activations into interpretable latents $a_{l,j}(\mathbf{x}_l)$ at each layer $l$.

We exclude features activating $>2\%$ on the pile-10k dataset to filter out general language patterns. From this filtered set, we apply complementary metrics suited to each mechanism's computational role. Throughout our analysis, $N^{\text{correct}}$ and $N^{\text{incorrect}}$ denote the total number of correct and incorrect code samples, respectively.

**Prediction directions** require sensitivity to confidence gradients. Code correctness manifests not as binary presence but as activation intensity differences between correct and incorrect samples. The t-statistic captures these graded signals while accounting for variance:

$$t_{l,j}^{\text{correct}} = \frac{\mu(a_{l,j}(\mathbf{x}_i^{\text{correct}})) - \mu(a_{l,j}(\mathbf{x}_i^{\text{incorrect}}))}{\sqrt{\frac{\sigma(a_{l,j}(\mathbf{x}_i^{\text{correct}}))^2}{N^{\text{correct}}} + \frac{\sigma(a_{l,j}(\mathbf{x}_i^{\text{incorrect}}))^2}{N^{\text{incorrect}}}}, \quad t_{l,j}^{\text{incorrect}} = \frac{\mu(a_{l,j}(\mathbf{x}_i^{\text{incorrect}})) - \mu(a_{l,j}(\mathbf{x}_i^{\text{correct}}))}{\sqrt{\frac{\sigma(a_{l,j}(\mathbf{x}_i^{\text{correct}}))^2}{N^{\text{correct}}} + \frac{\sigma(a_{l,j}(\mathbf{x}_i^{\text{incorrect}}))^2}{N^{\text{incorrect}}}}$$

$$(5)$$

where $\mu$ and $\sigma$ denote the mean and standard deviation of non-zero activations. This identifies features encoding model confidence about code correctness.

**Steering directions** demand categorical exclusivity for clean intervention. We first compute how frequently each feature activates:

$$f_{l,j}^{\text{correct}} = \frac{1}{N^{\text{correct}}} \sum_{i=1}^{N^{\text{correct}}} \mathbf{1}[a_{l,j}(\mathbf{x}_{l,i}^{\text{correct}}) > 0], \quad f_{l,j}^{\text{incorrect}} = \frac{1}{N^{\text{incorrect}}} \sum_{i=1}^{N^{\text{incorrect}}} \mathbf{1}[a_{l,j}(\mathbf{x}_{l,i}^{\text{incorrect}}) > 0]$$

$$(6)$$

Separation scores then measure exclusivity:

$$s_{l,j}^{\text{correct}} = f_{l,j}^{\text{correct}} - f_{l,j}^{\text{incorrect}}, \quad s_{l,j}^{\text{incorrect}} = f_{l,j}^{\text{incorrect}} - f_{l,j}^{\text{correct}} \tag{7}$$

High separation indicates switch-like features firing predominantly for one code type, enabling targeted corrections without affecting the opposite category.

Searching across all 26 model layers, we select features with maximum t-statistics for prediction and maximum separation scores for steering.

| Feature | Layer | Index | Metric | Used in |
|---|---|---|---|---|
| Correct Predicting | 16 | 14439 | t-stat: 5.086 | 5.1, 5.5 |
| Incorrect Predicting | 19 | 5441 | t-stat: 5.680 | 5.1, 5.5 |
| Correct Steering | 16 | 11225 | sep: 0.221 | 5.2, 5.3, 5.4, 5.5 |
| Incorrect Steering | 25 | 2853 | sep: 0.201 | 5.2, 5.3, 5.4, 5.5 |

Table 1: Key features identified for mechanistic analysis with their usage across sections.

## 5 MECHANISTIC ANALYSIS

### 5.1 DETECTION DIRECTIONS PREDICT ERRORS RELIABLY

Predictor directions reveal that models develop anomaly detectors rather than validity assessors. Despite similar AUROC scores ($\sim$0.6), incorrect-preferring features achieve F1=0.821 while correct-preferring features reach only 0.504.

Inspecting the top positive logits uncovers what these features detect. The incorrect-predicting feature activates on anomalous patterns such as null indicators, achieving 0.985 recall and 0.703 precision by detecting irregularities characteristic of errors. Unexpectedly, it also responds to foreign language tokens, providing empirical evidence that while SAEs decompose into sparse features, they do not fully solve polysemanticity. Correct-preferring features show worse specificity, activating on formatting tokens rather than semantic patterns. This produces extensive false positives as the feature mistakes well-formatted incorrect code for correct implementations. The metrics confirm this failure: while recall reaches 0.828 from detecting structured code, precision drops to 0.362 due to false positives, resulting in an F1 of 0.504 that shows the limitations of surface-level detection.

| POSITIVE LOGITS ⓘ | | POSITIVE LOGITS ⓘ | |
|---|---|---|---|
| سهل | 2.10 | none | 1.35 |
| سا | 1.38 | None | 1.28 |
| اسهل | 1.19 | none | 1.24 |
| \<eos\> | 1.13 | None | 1.21 |
| اساسا | 1.10 | SourceChecksum | 1.01 |
| اساسياً | 0.98 | NONE | 0.96 |
| اساساً | 0.80 | NONE | 0.94 |
| اسهلها | 0.77 | autorytatywna | 0.89 |
| \</h4\> | 0.74 | なし | 0.87 |
| اساسيات | 0.68 | لاسمء ء | 0.83 |

Figure 2: Top 10 tokens with the highest logit increases from predictor features. **Left:** Correct-predicting feature (L16-14439). **Right:** Incorrect-predicting feature (L19-5441).

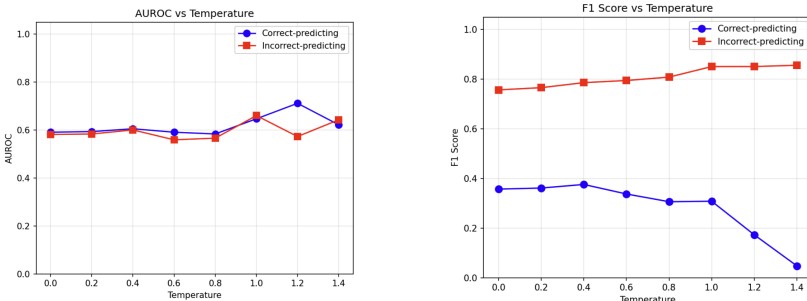

Figure 3: Temperature robustness analysis from T=0.0 to T=1.4. **Left:** AUROC scores. **Right:** F1 scores.

This anomaly-detection architecture remains stable. Temperature variations (0.0-1.4) leave error detection intact or improved (F1: 0.821→0.986), as anomalies remain detectable regardless of sampling randomness. Meanwhile, formatting-based correct detection degrades, confirming its superficial nature.

These findings reveal an asymmetry: models encode incorrect code as detectable anomalies but lack corresponding representations for correctness. While this prevents using these features as general confidence indicators, the reliable error detection (F1: 0.821) suggests practical utility as an alarm system, flagging generations that require review.

## 5.2 STEERING DIRECTIONS ACHIEVE MODEST CORRECTIONS

Transitioning from correlational identification to causal validation, we employ activation steering (Turner et al., 2023) to test whether separation-score directions influence code generation. Our intervention modifies the model's residual stream:

$$\mathbf{x}^{\text{steered}} = \mathbf{x} + \alpha \cdot \mathbf{W}_{\text{dec}}[j, :] \tag{8}$$

where $\mathbf{W}_{\text{dec}}[j, :]$ is the latent direction and $\alpha$ controls steering strength. We optimize steering coefficients through a two-phase search, where grid search (intervals of 10) identifies active ranges before the golden section search pinpoints optimal values. For correct steering, we maximize correction rate ($\alpha = 29$), while for incorrect steering we maximize $\frac{1}{2}(C_r + S)$ where $C_r$ denotes corruption rate and $S$ the mean Python token similarity percentage ($\alpha = 287$), preserving code structure while breaking functionality.

We employ three setups: baseline (no steering, hence 0% correction/corruption as code maintains initial state), control (feature L1-4801 with zero discrimination, coefficients $\alpha = 29$ for correct steering comparison and $\alpha = 287$ for incorrect steering comparison), and steering directions. One-tailed (greater) binomial testing validates two hypotheses: code correctness directions versus baseline tests

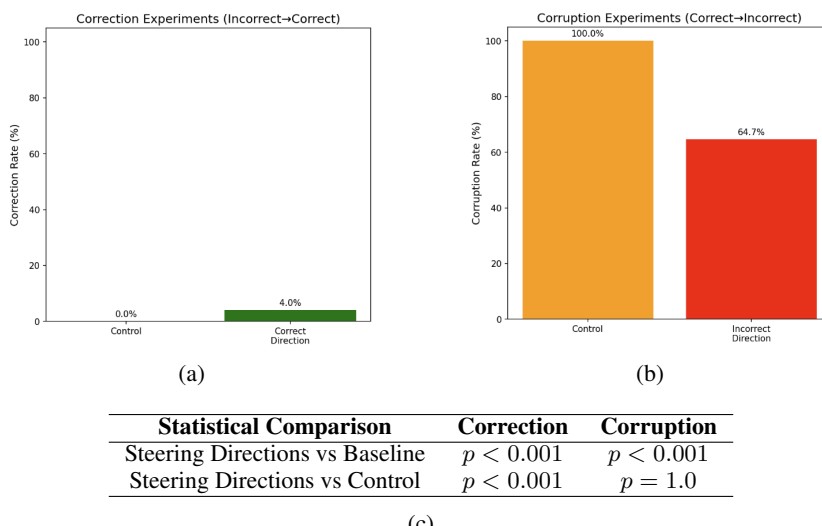

| Statistical Comparison | Correction | Corruption |
|---|---|---|
| Steering Directions vs Baseline | $p < 0.001$ | $p < 0.001$ |
| Steering Directions vs Control | $p < 0.001$ | $p = 1.0$ |

(c)

Figure 4: (a) Correction rates. (b) Corruption rates. (c) Statistical comparison using one-tailed (greater) binomial tests.

if steering has any effect, while code correctness directions versus control tests if our feature selection matters beyond random perturbation.

```python
# Before steering                          # Before steering
def char_frequency(string):               def volume_sphere(r):
    return dict.fromkeys(string, 0)            return (4/3)*3.141592653589793*r**3

# After steering                           # After steering
def char_frequency(string):               def volume_sphere(r):
    frequency = {}                            return 8888888888888888888...
    for char in string:
        if char in frequency:
            frequency[char] += 1
        else:
            frequency[char] = 1
    return frequency
```

Figure 5: **Left:** Correct-steering example. **Right:** Incorrect-steering example.

Steering interventions validate causal influence while revealing inherent tradeoffs. Correct-steering achieves 4.04% correction rate on initially incorrect code (p<0.001), with Figure 5 providing a concrete example. Yet this same intervention corrupts 14.66% of initially correct code. This degradation rate exceeds the correction rate nearly fourfold, suggesting selected steering rather than constant steering. Logit analysis reveals that correct-steering amplifies formatting tokens (spaces, tabs, comments), yet corrected code samples demonstrate semantic improvements, including bug fixes and algorithm implementations. The gap between formatting-related logits and semantic corrections illustrates why steering experiments are more informative than logit analysis alone.

Incorrect-steering's failure is more straightforward, with '8' token repetition in steered code (Figure 5), confirming that separation scores are ineffective at identifying incorrect features. This is further supported by Figure 6, where the tokens are predominantly variations of 'eight'. This failure pattern explains why the steering coefficient search algorithm settles on the comparatively large value of 287, nearly 10-fold larger than the correct steering coefficient of 29. Such large magnitudes mean that even control features produce a substantial impact when steered at this coefficient, explaining their complete degeneracy.

## 5.3 TEST CASES MATTER MORE THAN PROBLEM DESCRIPTIONS

Attention weight analysis reveals how steering interventions redistribute focus across prompt components. We extract attention weights from all heads at the steering layer (L16 for correct, L25

POSITIVE LOGITS ⑦

| | |
|---|---|
| | 0.61 |
|  | 0.59 |
| | 0.59 |
| | 0.58 |
| // | 0.57 |
| | 0.55 |
| | 0.55 |
| | 0.54 |
| | 0.53 |
| | 0.51 |

POSITIVE LOGITS ⑦

| | |
|---|---|
| 8 | 2.03 |
| eight | 1.59 |
| Eight | 1.55 |
| Eighth | 1.46 |
| eighth | 1.44 |
| EIGHT | 1.37 |
| 8 | 1.33 |
| /\ | 1.31 |
| Eight | 1.29 |
| delapan | 1.29 |

Figure 6: Top 10 tokens with the highest logit increases from steering features. **Left:** Correct-steering feature (L16-11225). **Right:** Incorrect-steering feature (L25-2853).

for incorrect) at the final instruction token, sum attention within three prompt sections, normalize each section to percentages of total attention, and then compute percentage point changes between baseline and steered conditions.

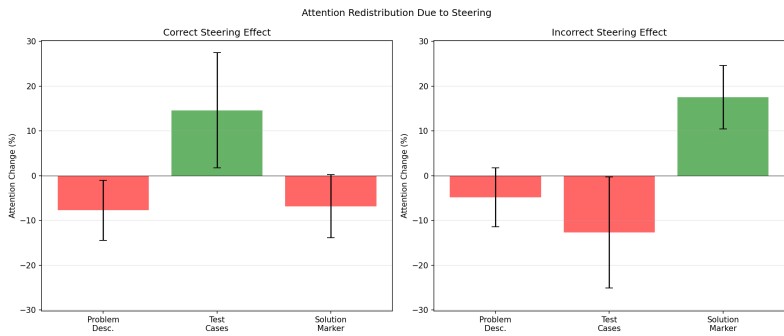

Figure 7: Percentage point changes in attention to problem descriptions, test cases, and code initiator under steering interventions.

Test cases exhibit the largest differential at 27.29 points between correct (+14.60) and incorrect (-12.69) steering. Problem descriptions decrease regardless of steering direction, never exceeding an 8-point reduction. The code initiator, a prompt artifact less relevant than problem descriptions and test cases, receives increased attention (+17.54) under incorrect-steering, confirming these features disrupt information processing. These patterns demonstrate that successful code generation depends on attending to test cases rather than problem descriptions. This mechanism suggests that prompting strategies should prioritize concrete test examples over detailed problem descriptions.

### 5.4 CORRECT DIRECTIONS PROVE NECESSARY FOR GENERATION

Weight orthogonalization permanently modifies every matrix writing to the residual stream:

$$\mathbf{W}_{\text{out}}^{\text{new}} \leftarrow \mathbf{W}_{\text{out}} - \mathbf{W}_{\text{out}}\mathbf{d}^T\mathbf{d} \tag{9}$$

following Arditi et al. (2024), this prevents the model from writing the specified direction $\mathbf{d}$. We employ three setups: baseline (no orthogonalization, hence 0% correction/corruption as code maintains initial state), control (feature L1-4801 with zero discrimination), and steering directions.

Correct orthogonalization corrupts 83.6% of initially correct solutions, compared to only 19.0% for control features (p<0.001), a 4.4-fold difference. As shown in Figure 8(a), functional code degrades to comments or empty strings when these features are removed. The model retains task knowledge (evidenced by relevant comments) but cannot produce executable code. This demonstrates that correct-steering features are necessary for code generation.

Incorrect orthogonalization shows contrasting results. We expected removing incorrect steering directions would reduce errors, but achieved only a 2.2% correction rate, below control features at

5.5%. This indicates our failure to identify effective incorrect-preferring directions, consistent with Section 5.2, where separation scores proved ineffective for finding incorrect features.

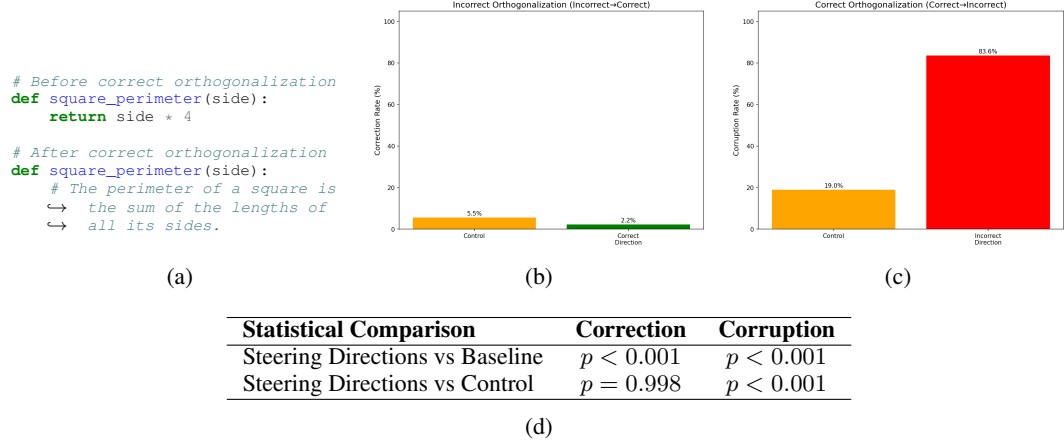

(a)  (b)  (c)

| Statistical Comparison | Correction | Corruption |
|---|---|---|
| Steering Directions vs Baseline | $p < 0.001$ | $p < 0.001$ |
| Steering Directions vs Control | $p = 0.998$ | $p < 0.001$ |

(d)

Figure 8: (a) Code example before and after correct orthogonalization. (b) Incorrect orthogonalization correction rates. (c) Correct orthogonalization corruption rates. (d) Statistical comparison using one-tailed (greater) binomial tests.

## 5.5 MECHANISMS PERSIST FROM BASE TO CHAT MODELS

GemmaScope SAEs were trained exclusively on the base model using pre-training data, yet SAE-derived directions retain their effectiveness in the instruction-tuned model. This persistence occurs despite instruction-tuning improving baseline performance from 29.9% to 38.4% pass rate on MBPP.

Error detection remains reliable across both models. The incorrect-preferring feature (L19-5441) achieves F1=0.821 in the base model and F1=0.772 after instruction-tuning. Both models maintain F1>0.75 for incorrect prediction, preserving the reliability threshold established in Section 5.1.

Steering interventions retain statistical significance. Correct-steering achieves 4.04% correction rate in the base model (p<0.001) and 2.93% in the instruction-tuned model (p<0.001). While the rate decreases, both models demonstrate statistically significant correction ability using identical features and coefficients.

These results indicate code correctness mechanisms learned during pre-training persist through instruction-tuning. Rather than developing new mechanisms, fine-tuning appears to refine existing representations while maintaining their fundamental structure. This persistence enables SAEs trained on base models to identify causally relevant features in their instruction-tuned counterparts.

## 6 CONCLUSIONS

Using sparse autoencoders, we identified and characterized code correctness directions in LLM representations, finding that predictor directions reliably detect incorrect code (F1: 0.821) while steering directions achieve corrections with inherent tradeoffs (4.04% fixed, 14.66% corrupted). Mechanistically, we demonstrated that successful code generation depends on attending to test cases rather than problem descriptions. Notably, incorrect-predicting and correct-steering directions identified in base models retain their effectiveness after instruction-tuning, suggesting code correctness mechanisms learned during pre-training are repurposed during fine-tuning. These mechanistic insights suggest practical applications: prompting strategies should prioritize test examples over elaborate problem descriptions, predictor directions can serve as error alarms for developer review, and these same predictors can guide selective steering, intervening only when errors are anticipated to prevent the 14.66% corruption rate from constant steering. This work advances the mechanistic interpretability of code processing in LLMs, revealing how models internally represent and process code correctness.

## 7    LLM USAGE

We used Claude (Anthropic) to assist with multiple aspects of this work. For implementation, we used Claude Code to generate analysis code for SAE decomposition, steering interventions, and attention analysis based on our specifications, with all code being manually reviewed, tested, and validated. For manuscript preparation, Claude assisted with initial draft generation, which we subsequently revised and refined. For the literature review, we used Claude to identify potentially relevant papers, though all citations were independently verified for relevance and accuracy. All experimental design, analysis, interpretation of results, and scientific conclusions are our own. We take full responsibility for all content in this paper.

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
