# OpenReview forum: "Mechanistic Interpretability of Code Correctness in LLMs via Sparse Autoencoders"
_ICLR.cc/2026/Conference — Submitted to ICLR 2026_

### Official Review · Reviewer_jwL5 · 2025-10-22

**Soundness:** 2
**Presentation:** 2
**Contribution:** 2
**Rating:** 4
**Confidence:** 4

**Summary:**

This paper uses layer-specific sparse autoencoders on Gemma-2-2B’s residual stream to identify “predictor” directions that distinguish correct vs incorrect code generations and “steering” directions that, when added at inference, bias the model’s generation. The authors produce key findings on coding benchmark MBPP, the best “incorrect-predicting” feature serves as a modest error alarm (F1 = 0.82), while the “correct-predicting” feature is weak (F1 = 0.504). Steering with the “correctness” direction repairs ~4.0% of initially wrong solutions but also corrupts ~14.7% of initially correct ones. The authors also perform an attention analysis showing steering shifts focus towards the unit tests. The authors also suggest that the correct directions are causal for code correctness behaviour as orthogonalizing a “correct” direction degrades ~83.6% of previously correct solutions vs. ~19% for a neutral control. The authors found that these directions largely transfer from the base model to a chat-tuned variant, indicating some robustness of the discovered mechanisms.

**Strengths:**

Novelty: This is the first mechanistic interpretability study specifically aimed at code correctness in LLMs, offering fresh insights into a relatively unexplored question that prompts opportunities for follow-up directions in the research community.

Transfer Signal: Some of the discovered directions carry over from base to chat-tuned model, hinting these are not pure overfits to one checkpoint.

**Weaknesses:**

**Methodology Ambiguity:**
The paper’s high-level recipe is clear, but many operational details are missing or scattered, which makes replication and interpretation difficult. In particular, several core steps are not specified precisely enough to rule out confounds or to let others reproduce the results end-to-end:
- Definition of S (similarity): “Mean Python token similarity %” is undefined (tokenizer, reference, surface vs AST notion of similarity, averaging window). Clarifying this would strengthen the understanding.
- Evaluation bookkeeping: How “initially correct” is labeled, accuracy, and sample counts per condition. Clarifying this would strengthen the claims.
- Control condition: The neutral control feature has not been explored and a single control is weak to support specificity claims.

**Feature-Selection Landscape:**
The current search/selection feels under-characterized. Please add a compact Feature-Selection Landscape to show the chosen directions are genuine outliers rather than hand-picked. These additions make selection auditable at a glance, demonstrate how exceptional the chosen features are, and strengthen this paper’s evidence.

**Single Benchmark Scope:**
All results are on MBPP (short Python tasks). To support this paper
claims about code correctness more broadly, it would help to include at least one additional benchmark (HumanEval, APPS) and, ideally, also non-Python. Showing that the key findings (e.g. stronger error alarm, test-focused attention, and orthogonalization effects) transfer across datasets/languages will materially improve generality and impact.

**Questions:**

Small Comment:
It would be great to reference Figure 7 in the text.

---

### Official Review · Reviewer_7JAK · 2025-10-31

**Soundness:** 3
**Presentation:** 2
**Contribution:** 2
**Rating:** 6
**Confidence:** 4

**Summary:**

The authors apply previously developed methods to analyze LLM performance in code generation tasks. This is a field that has seen widespread adoption as in both academia and industry the share LLM generated code is increasing.  The authors applied sparse autoencoders on the residual streams of each layer to disentangle super-positioned representations and then detect the most impactful latent features. These were analyzed and intervened upon through a steering method. The authors confirmed previous results, that LLMs display anomaly detection mechanisms and not validity assessments. Furthermore, they found that applying steering directions to identified latent features leads to a trade-off between accurate code corrections and corrupting previously well generated code.

**Strengths:**

a. Important and Timely Research Question
	•	Addresses critical need for understanding LLM code generation reliability as these models enter industrial deployment
	•	Focuses on mechanistic interpretability, which is essential for trustworthy AI systems

b. Novel Methodological Approach
	•	Creative application of SAEs to decompose code correctness representations into interpretable features
	•	Addresses the superposition problem that compresses high-dimensional features into lower dimensions
	•	Uses multiple complementary analysis techniques (steering, attention, weight orthogonalization)

c. Well-Structured Experimental Design
	•	Systematic approach using t-statistics and separation scores
	•	Demonstrates both predictive capabilities and correction mechanisms
	•	Acknowledges trade-offs in the findings (error fixing vs. correct code preservation)

**Weaknesses:**

a. All of the analysis was performed on the latent features with highest metric as displayed in Table 1. Do the conclusions hold for say the top-5 to -10 latent features? Are the metric values for these features outliers compared to other features? Some further details in this aspect would significantly enhance the robustness of results.

b. There is some lack of clarity when reading the paper, particularly if unfamiliar with the methods developed in Ferrando et al., 2024 and Marks & Tegmark, 2023. There was some effort to mention details of the methods, although not enough was included to fully grasp the methods (In some cases it could better to simply guide the reader to the original paper). Some aspects that could be addressed include:

1. Clarify indexing of equations 5-7

2. Clarify the use of the identified latent feature for classification through thresholding, as understanding the appearance of AUROC and F1 scores was not clear.

3. Personally, the usage of “feature” for the encoded position in the SAE can be error inducing as the word is more commonly used for input “features”. Possibly using “latents” (as in Ferrando et al., 2024) could be better.

4. Why was feature L1-4801 used as control?

5. Some further clarification on how the steering strength parameter was obtained could be added as supplementary material.

c. The values plotted in Figure 3 generally do not seem to match the values mentioned in the text. The F1 value for incorrect-predicting at T=0 is below 0.8 while the reference value in the text is 0.821, and in line 246 the improvement 0.821->0.986 is mentioned whereas the values seem to lie in the [0.76, 0.83] interval.

d. The key suggestion given by the authors from the obtained results, is to include more test cases when prompting. While this is easily applicable in the used dataset for simple methods, it is often not so simple when developing methods in real scenarios. It would have a larger impact on SOTA if suggestions on improving LLMs to achieve validity assessment mechanisms.


Minor suggestion:

a. MBPP is not defined in line 56, later defined in line 140

b. The link of reference Elhage et al. 2022 is incomplete, while the text is present the link only goes to “/toy”.

**Questions:**

1. Overall while the authors did a good job it misses the core question of how does the issue of entanglement be solved.

2. Methodological Gaps
	•	No clear guarantee of SAE uniqueness - how do authors ensure SAEs avoid the entanglement problem they claim to solve?
	•	Mean deviation process in prediction direction needs better motivation and explanation
	•	Missing details on how different values in Lines 261-265 are obtained

3. Conceptual and Terminological Issues
	•	"Causal influence" (L306) would be more accurately termed "intervention"
	•	Introduction could better connect SAE methodology symbols to the actual methodology section
	•	Some claims about superposition and feature compression need stronger theoretical grounding

4. Incomplete Analysis
	•	Limited discussion of potential failure modes or limitations of the SAE approach
	•	Insufficient exploration of how findings generalize across different code types and complexity levels

---

### Official Review · Reviewer_vRko · 2025-11-04

**Soundness:** 2
**Presentation:** 2
**Contribution:** 2
**Rating:** 4
**Confidence:** 3

**Summary:**

This paper investigates the "code correctness" mechanism of Large Language Models (LLMs) in code generation tasks, motivated by the widespread application of AI code generation but with inherent reliability risks, especially in high-risk domains.
The authors decompose the internal representation of LLMs using a Sparse Autoencoder (SAE) to identify directions relevant to code correctness: detected directions are used to predict errors, while manipulated directions can be used to correct them.
The paper systematically analyzes the performance, attention distribution, and persistence of these mechanisms after model fine-tuning, finding that test cases are more critical than problem descriptions, and offers implications for practical development processes, such as improved prompt design and automatic error alerts.

**Strengths:**

- This study elucidates the mechanism of using sparse autoencoders in code generation models, overcoming the interpretability challenge caused by feature superposition.

- It covers multiple aspects, including detection, manipulation, attention distribution, and weight orthogonalization, with rigorous experimental design and thorough statistical testing.

- It proposes superior code suggestion strategies, error alerting mechanisms, and targeted model intervention recommendations, demonstrating practical engineering value.

- It proves that the correctness mechanism in the pre-trained model remains effective after fine-tuning, possessing both theoretical significance and practical applicability.

**Weaknesses:**

- The correction rate was only 4.04%, while erroneous intervention resulted in 14.66% of correct code being corrupted, indicating a high risk in the practical application of the manipulation direction and insufficient exploration of how to reduce side effects.

- Although key experiments were conducted, the experiments were not comprehensive enough: (1) All experiments were based on Gemma-2 and MBPP, failing to cover other models or more complex code scenarios, and the generalizability of the results needs further verification. (2) Negative mechanism identification failed: the "erroneous code" direction was not effectively identified, and the related experimental results and analysis were relatively weak.

- The pipelines relied on closed-source large models: Although manually reviewed, some experiments and literature searches relied on Claude, which may affect independence.

- Missing some references:
  - Sparse Autoencoders Find Highly Interpretable Features in Language Models
  - A Survey on Sparse Autoencoders: Interpreting the Internal Mechanisms of Large Language Models
  - Sparse Autoencoders Find Highly Interpretable Features in Language Models

**Questions:**

- Can the features decomposed by a sparse autoencoder be generalized to other coding tasks (such as multi-language, multi-file projects)?

- How can the manipulation direction be optimized to reduce the rate of breaking correct code and make it more suitable for practical deployment?

- In actual development processes, how can the detection direction be combined with existing static analysis tools to improve code quality?

- The paper failed to effectively identify the "erroneous code" direction, are there alternative methods or future plans?

---

### Official Review · Reviewer_RXZd · 2025-11-04

**Soundness:** 3
**Presentation:** 3
**Contribution:** 3
**Rating:** 2
**Confidence:** 4

**Summary:**

This paper applies sparse autoencoders (SAEs) to decompose LLM representations and identify directions corresponding to code correctness in the Gemma-2-2b model. The authors discover two distinct mechanisms: detection directions that reliably predict incorrect code (F1: 0.821) but fail as confidence indicators for correct code (F1: 0.504), and steering directions that achieve modest corrections (4.04% of errors fixed) while corrupting 14.66% of initially correct code. Through mechanistic analysis including activation steering, attention weight analysis, and weight orthogonalization, the authors demonstrate that successful code generation depends primarily on attending to test cases rather than problem descriptions, and that correct-steering directions are causally necessary for code generation. The work represents the first application of SAEs to address superposition in code representations and suggests practical applications including using predictor directions as error alarms and applying selective rather than constant steering interventions.

**Strengths:**

- First use of SAEs to decompose code correctness representations, extending entity recognition frameworks to a new domain with appropriate adaptations.
- The paper employs multiple validation techniques (steering, attention analysis, weight orthogonalization) that provide converging evidence for causal mechanisms rather than mere correlations.
- The discovery that models encode incorrect code as detectable anomalies but lack corresponding representations for correctness reveals fundamental insights about how LLMs represent code validity.
- Demonstrating that code correctness mechanisms persist from base to chat models (F1: 0.821 → 0.772 for error detection) suggests interpretability methods can generalize across training stages.

**Weaknesses:**

- The entire analysis is conducted on a single model (Gemma-2-2b, 2B parameters) and single benchmark (MBPP). This raises serious questions about whether findings generalize to other model families, sizes, or programming tasks (e.g., HumanEval, APPS, more complex algorithms). The authors should at minimum validate key findings on one additional model and benchmark.
- The 4.04% correction rate is overshadowed by the 14.66% corruption rate which is nearly 4-fold difference. While the authors acknowledge this necessitates "selective steering," they don't demonstrate or evaluate such a selective approach. The practical utility remains unclear without showing that combining detection and steering actually improves overall performance.
- The incorrect steering direction identified via separation scores produces only repetitive '8' tokens and achieves merely 2.2% corrections (below control at 5.5%). This represents a fundamental failure of the separation score methodology for identifying incorrect features.
- The discovery that the incorrect-predicting feature activates on both anomalous code patterns AND foreign language tokens directly contradicts the SAE promise of monosemantic features. The paper dismisses this as "empirical evidence that SAEs do not fully solve polysemanticity" but doesn't adequately discuss implications for the reliability of their findings or alternative approaches.

**Questions:**

- Refer to weaknesses
- You exclude features activating >2% on pile-10k. How sensitive are results to this threshold?
- Do detection and steering directions work differently for different error types (syntax errors, logic errors)? Breaking down performance by error category would be valuable.

---

### Meta-Review · Area_Chair_N7vS · 2026-01-07

**Summary:**

Across reviews, the work is viewed as a timely and potentially novel mechanistic interpretability study that uses sparse autoencoders to isolate representation directions correlated with code correctness, supported by multiple “triangulating” analyses (steering, attention shifts, orthogonalization/causal tests) and some evidence of transfer from a base to a chat-tuned model. However, reviewers’ suggested decision is primarily driven by (i) limited scope (single small model and single benchmark), (ii) unclear practical benefit of the intervention (low fix rate paired with substantially higher corruption rate, without a demonstrated selective strategy), and (iii) methodological/clarity gaps that limit reproducibility and weaken confidence in claims (feature-selection auditability, missing operational details, and some reported-number/figure inconsistencies). With three reviewers at/below the acceptance threshold and one slightly above, the balance of concerns supports rejection in the current form.

**Reviewer Concerns:**

Unaddressed concerns due to no rebuttal:
- Multiple reviewers emphasize that results are confined to one model (Gemma-2-2B) and one benchmark (MBPP) and request validation on at least one additional model and dataset (e.g., HumanEval/APPS, broader tasks/languages).
- Reviewers agree the ~4% correction vs ~15% corruption trade-off undermines practical usefulness, especially without demonstrating selective steering or a combined detection→intervention pipeline that yields net gains.
- Several highlight that the “incorrect/erroneous” direction identification appears to fail (e.g., degenerate outputs, below-control performance), calling into question parts of the feature-selection methodology.
- One reviewer flags that a feature activating for both anomalous code and foreign-language tokens weakens monosemantic claims and asks for deeper discussion of implications and reliability.
- Requests include clearer definitions (e.g., similarity metric and labeling/bookkeeping), stronger controls, an auditable feature-selection landscape (to show directions are genuine outliers rather than potentially cherry-picked), and resolving apparent figure/text mismatches.
- Missing references and some concern about reliance on closed-source tools for parts of the pipeline.

**Reviewer Scores:**

- RXZd (2 → ~2): Likely unchanged; core concerns are generalization and intervention trade-off, plus methodological issues.
- vRko (4 → ~4): Likely unchanged; sees promise but wants broader validation and safer/more effective manipulation.
- 7JAK (6 → ~5–6): Might slightly decrease if pressed on scope/clarity/number inconsistencies, but could also remain marginally positive due to perceived timeliness and multi-pronged analysis.
- jwL5 (4 → ~4): Likely unchanged; main issues are missing operational detail, weak controls/selection auditability, and single-benchmark scope.

---

### Decision · Program_Chairs · 2026-01-26

Reject